# Protective effect of pre-existing natural immunity in a nonhuman primate reinfection model of congenital cytomegalovirus infection

**Matilda J. Moström**[1], **Shan Yu**[1], **Dollnovan Tran**[1], **Frances M. Saccoccio**[2], **Cyril J. Versoza**[3], **Daniel Malouli**[4], **Anne Mirza**[5], **Sarah Valencia**[2], **Margaret Gilbert**[1], **Robert V. Blair**[1], **Scott Hansen**[4], **Peter Barry**[6], **Klaus Früh**[4], **Jeffrey D. Jensen**[3], **Susanne P. Pfeifer**[3], **Timothy F. Kowalik**[5], **Sallie R. Permar**[2,7]*, **Amitinder Kaur**[1]*

1 Tulane National Primate Research Center, Tulane University, Covington, Louisiana, United States of America, 2 Duke Human Vaccine Institute, Duke University, Durham, North Carolina, United States of America, 3 Center for Evolution & Medicine, School of Life Sciences, Arizona State University, Tempe, Arizona, United States of America, 4 Oregon Health and Sciences University, Beaverton, Oregon, United States of America, 5 University of Massachusetts Chan Medical School, Worcester, Massachusetts, United States of America, 6 University of California, Davis, California, United States of America, 7 Weill Cornell Medicine, New York, New York State, United States of America

* sap4017@med.cornell.edu (SP); akaur@tulane.edu (AK)

**Data Availability Statement:** Raw data has been uploaded as supplementary information in an excel

## Abstract

Congenital cytomegalovirus (cCMV) is the leading infectious cause of neurologic defects in newborns with particularly severe sequelae in the setting of primary CMV infection in the first trimester of pregnancy. The majority of cCMV cases worldwide occur after non-primary infection in CMV-seropositive women; yet the extent to which pre-existing natural CMV-specific immunity protects against CMV reinfection or reactivation during pregnancy remains ill-defined. We previously reported on a novel nonhuman primate model of cCMV in rhesus macaques where 100% placental transmission and 83% fetal loss were seen in CD4+ T lymphocyte-depleted rhesus CMV (RhCMV)-seronegative dams after primary RhCMV infection. To investigate the protective effect of preconception maternal immunity, we performed reinfection studies in CD4+ T lymphocyte-depleted RhCMV-seropositive dams inoculated in late first / early second trimester gestation with RhCMV strains 180.92 (n = 2), or RhCMV UCD52 and FL-RhCMVΔRh13.1/SIVgag, a wild-type-like RhCMV clone with SIVgag inserted as an immunological marker, administered separately (n = 3). An early transient increase in circulating monocytes followed by boosting of the pre-existing RhCMV-specific CD8+ T lymphocyte and antibody response was observed in the reinfected dams but not in control CD4+ T lymphocyte-depleted dams. Emergence of SIV Gag-specific CD8+ T lymphocyte responses in macaques inoculated with the FL-RhCMVΔRh13.1/SIVgag virus confirmed reinfection. Placental transmission was detected in only one of five reinfected dams and there were no adverse fetal sequelae. Viral whole genome, short-read, deep sequencing analysis confirmed transmission of both reinfection RhCMV strains across the placenta with ~30% corresponding to FL-RhCMVΔRh13.1/SIVgag and ~70% to RhCMV UCD52, consistent with the mixed human CMV infections reported in infants with cCMV. Our data showing reduced placental transmission and absence of fetal loss after non-primary as

file titled "S1 Data". Each tab of the excel spreadsheet has the data for the indicated figures.

**Funding:** This work was supported by NIH grants DP2HD075699, P01AI129859, and P51OD011104. The funders had no role in study design, data collection and analysis, decision to publish, or preparation of the manuscript.

**Competing interests:** "I have read the journal's policy and the authors of this manuscript have the following competing interests: Sallie Permar serves as a consultant to GSK, Moderna, Merck, Pfizer, Hoopika, and Dynavax vaccine programs, as well as leading a sponsored research program with Moderna and Merck. Oregon Health Sciences University (OHSU), Klaus Früh, Daniel Malouli and Scott Hansen have a substantial financial interest in Vir Biotechnology, Inc. a company that may have a commercial interest in the results of this research and technology. Klaus Früh, Daniel Malouli and Scott Hansen are co-inventors of several patents licensed to Vir Biotechnology. Klaus Früh and Scott Hansen are also consultants to Vir Biotechnology, Inc. All potential conflicts of interest have been reviewed and managed by OHSU. All other authors report no potential conflicts"

opposed to primary infection in CD4+ T lymphocyte-depleted dams indicates that preconception maternal CMV-specific CD8+ T lymphocyte and/or humoral immunity can protect against cCMV infection.

## Author summary

Globally, pregnancies in CMV-seropositive women account for the majority of cases of congenital CMV infection but the immune responses needed for protection against placental transmission in mothers with non-primary infection remains unknown. Recently, we developed a nonhuman primate model of primary rhesus CMV (RhCMV) infection in which placental transmission and fetal loss occurred in RhCMV-seronegative CD4+ T lymphocyte-depleted macaques. By conducting similar studies in RhCMV-seropositive dams, we demonstrated the protective effect of pre-existing natural CMV-specific CD8+ T lymphocytes and humoral immunity against congenital CMV after reinfection. A 5-fold reduction in congenital transmission and complete protection against fetal loss was observed in dams with pre-existing immunity compared to primary CMV in this model. Our study is the first formal demonstration in a relevant model of human congenital CMV that natural pre-existing CMV-specific maternal immunity can limit congenital CMV transmission and its sequelae. The nonhuman primate model of non-primary congenital CMV will be especially relevant to studying immune requirements of a maternal vaccine for women in high CMV seroprevalence areas at risk of repeated CMV reinfections during pregnancy.

## Introduction

Human cytomegalovirus (CMV) is a betaherpesvirus that results in lifelong persistent infection. While infection in immunocompetent hosts is typically asymptomatic, CMV causes life-threatening illness in immunosuppressed hosts such as transplant recipients and individuals with untreated HIV infection. CMV is also the most common cause of congenital infection in newborns with severe neurological sequelae resulting from primary infection in CMV-seronegative women in the first trimester of pregnancy [1]. Congenital CMV (cCMV) can also follow non-primary infection in pregnant CMV-seropositive women. Non-primary infection due to either reactivation of endogenous latent CMV or reinfection with diverse CMV strains is the most common cause of human cCMV cases worldwide occurring in regions with high CMV seroprevalence rates among women of reproductive age [2–6]. Even in the United States, three-quarters of cCMV infections reported between 1988–1994 were attributed to non-primary infection [7]. Consequently, the development of a maternal vaccine to prevent cCMV has been a tier 1 priority of the National Institute of Medicine for two decades but has faced several challenges [8]. A major barrier to vaccine development is that the immune determinants of protection against vertical transmission, particularly after non-primary infection, remain elusive. Even though natural CMV infection induces a robust and durable humoral and cellular immune response, pre-existing immunity is not sufficient to prevent reinfections or cCMV. The majority of cCMV transmissions worldwide occur in regions with high maternal seroprevalence rates and account for >500,000 cCMV infant births per year despite "low" transmission rates of 1–2% [9]. Moreover, in high CMV seroprevalence regions the rate of CMV exposure and reinfection during pregnancy is up to 10 times higher than that of CMV-

seronegative women increasing the prevalence of non-primary cCMV [10]. To have world-wide relevance, an effective CMV vaccine will need to prevent both primary and non-primary cCMV. Insight in to the role of pre-existing natural immunity in limiting congenital CMV transmission and its protective components are thus of paramount importance for the rational design of an effective maternal vaccine against cCMV.

Several studies have reported a reduced fetal infection rate after non-primary compared to primary CMV infection. A meta-analysis of epidemiologic studies conducted between 1966 and 2006 revealed that the rate of intra-uterine CMV transmission after primary infection during pregnancy was 32%, as opposed to 1.4% after non-primary infection [11]. In a prospective study of 2,378 woman/newborn pairs with maternal CMV serology at first visit and CMV screening of newborns at birth, the risk of maternal-fetal transmission after primary infection was 4-fold higher compared to non-primary infection [12]. A similar reduction in transmission after non-primary infection has been reported from cCMV cohorts in Italy and Brazil where maternal primary and non-primary infections were monitored by serial serology [13, 14]. It is worth noting that in contrast to transmission, the symptomatology and severity of cCMV with respect to sensorineural hearing loss and neurodevelopmental anomalies can be comparable in non-primary and primary infection [15–17]. These data suggest that the mechanisms protecting against vertical transmission may be different from those determining the outcome of fetal infection once transmission has occurred. Elucidating the determinants of protection against acquisition and disease severity, particularly the role of pre-conceptional immunity in CMV-seropositive pregnant women, is of paramount importance for the development of a maternal vaccine effective against both primary and non-primary cCMV infection [14, 18]. In this regard, the rhesus macaque CMV reinfection pregnancy model has the potential to address some of the unanswered questions pertaining to non-primary cCMV, including the rate of placental transmission, protective effects of pre-existent natural immunity, and the host response to reinfection.

The rhesus macaque animal model offers several translational benefits for studying the pathogenesis and immunology of human CMV (HCMV) infection. The rhesus CMV (RhCMV) genome is closely related to HCMV and the natural history and biology of RhCMV infection in rhesus macaques bears important similarities to HCMV infection [19–21]. Natural RhCMV infection is widespread in colony-bred rhesus macaques; near 100% seroprevalence rates are reached within a year of birth with natural acquisition occurring as a result of horizontal transmission from close contact with infectious body fluids containing shed virus [22–24]. Thus, breeding rhesus macaques have been CMV-seropositive for at least 2–3 years before reaching the age of sexual maturity. This mimics the natural history and infection profile of HCMV in regions of the world where CMV-seroprevalence in women of child-bearing age exceeds 95% as a result of primary infections occurring in childhood [18]. The rhesus macaque model of RhCMV infection is thus well suited for investigating immune determinants of protection against non-primary cCMV infection. We previously reported on a placental transmission model of primary cCMV infection in rhesus macaques [25]. In this model, we observed 100% vertical transmission and 83% fetal loss in CD4+ T lymphocyte-depleted CMV-seronegative macaques infected with RhCMV in late first / early second trimester gestation, with protection conferred by passive infusion of high potency anti-CMV antibodies prior to infection [25, 26]. Here we report the first instance of cCMV in CMV-seropositive rhesus macaques, representing a nonhuman primate (NHP) non-primary cCMV model.

Similar to our primary infection studies, pregnant macaques at late first / early second trimester gestation were subjected to CD4+ T lymphocyte depletion and then inoculated with RhCMV virus strains. The difference here was that the experiments were performed in CMV-seropositive macaques with pre-existing naturally acquired RhCMV-specific immunity that

were then experimentally inoculated with RhCMV to simulate reinfection during pregnancy. In a subset of animals we used FL-RhCMVΔRh13.1/SIV*gag*, a wild-type-like RhCMV clone with *SIVgag* inserted as an immunological marker [27], for experimental reinfection to enable *in vivo* tracking of the reinfection virus and distinguish it from reactivation of endogenous RhCMV. In contrast to primary RhCMV infection, reinfection of CD4+ T lymphocyte-depleted RhCMV-seropositive dams resulted in reduced placental transmission and complete protection against fetal loss, accompanied by a robust innate immune response and boosting of pre-existent CMV-specific immunity. Our study demonstrates the feasibility of using a NHP animal model to study non-primary cCMV infection and demonstrates the protective effect of pre-existing natural CMV-specific CD8+ T lymphocytes and humoral immunity in a biologically relevant model of human cCMV infection.

## Results

### Reinfection model of congenital CMV in rhesus macaques

To investigate the protective effect of natural pre-existing CMV-specific immunity against placental transmission, we applied the study design of our previously published CD4+ T lymphocyte depletion primary cCMV infection model [25]. We repeated the study design but substituted primary infection studies in RhCMV-seronegative macaques with non-primary reinfection studies in CD4+ T lymphocyte-depleted RhCMV-seropositive macaques with naturally acquired RhCMV infection. Colony-bred rhesus macaques have high RhCMV seroprevalence rates with natural acquisition of CMV by one year of age and thus have established pre-existing RhCMV-specific immunity for at least two years before reaching sexual maturity at 3–4 years of age [22, 24]. The reinfection experiments were performed after CD4+ T lymphocyte depletion using the same dose and timeframe of infection as in the primary infection model which had yielded 100% placental transmission. Hence, comparison of placental transmission in CD4+ T lymphocyte-depleted macaques with primary RhCMV infection and RhCMV reinfection enabled an evaluation of the protective effect of pre-existing natural RhCMV-specific CD8+ T lymphocyte and humoral immunity in RhCMV-seropositive dams.

Five RhCMV-seropositive dams were enrolled in the "RhCMV-seropositive Reinfection" group and three RhCMV-seropositive dams were enrolled in the "RhCMV-seropositive Control" group to control for endogenous RhCMV reactivation without reinfection (**Fig 1A**, **Table 1**). All dams were enrolled in the first trimester of pregnancy and subjected to *in vivo* CD4+ T lymphocyte depletion at late first trimester / early second trimester between 49 to 59 gestation days, similar to the historical control group of CD4+ T lymphocyte depletion and primary RhCMV infection (**Table 1**). One week after administration of the CD4+ T lymphocyte depleting antibody, the reinfection group dams were inoculated with different RhCMV strains. Two RhCMV-seropositive dams (274–05 and 292–09) received an intravenous inoculation of the fibroblast-passaged RhCMV strain 180.92 at $2x10^6$ $TCID_{50}$ and were followed until natural birth at 152 and 164 gestational days, respectively (**Table 1**, **Figs 1A and 2A**). RhCMV 180.92 is known to be a mixed virus; the UL/b' region of the dominant sequenced variant is truncated with a >5kb deleted segment that retains an intact UL128-UL131 locus [28]. It has been successfully used in experimental inoculation studies in RhCMV-seronegative macaques resulting in widespread viral dissemination in SIV-infected macaques [29] and placental RhCMV transmission in a CD4+ T-lymphocyte-depleted dam [25]. However, a minor wild-type like variant with a nontruncated UL/b' region rapidly emerges following experimental infection with RhCMV 180.92 and becomes the dominant strain *in vivo* in tissues and circulation [29]. In contrast, RhCMV strains UCD52 and UCD59 derived from limited passage of natural RhCMV isolates on monkey kidney epithelial cells, retain a full-length wild-type like

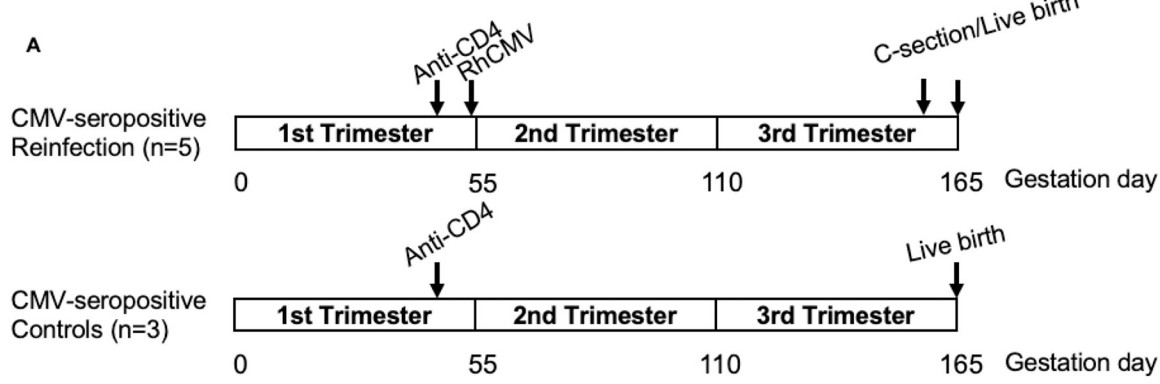

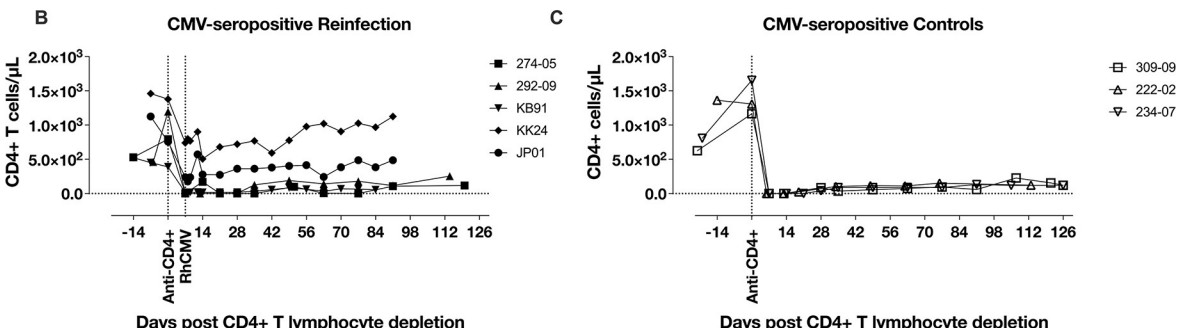

**Fig 1. Study design and kinetics of CD4+ T lymphocyte depletion in experimental groups.** (A) Schematic of study design of cCMV transmission in pregnant CMV-seropositive rhesus macaques. (B-C) Peripheral blood CD4+ T lymphocyte counts following anti-CD4 antibody administration in (B) CMV-seropositive Reinfection group ($n$ = 5); and (C) CMV-seropositive Control group ($n$ = 3).

UL/b' region of the RhCMV genome and are disseminated and shed into body fluids following experimental inoculation [30, 31]. Three dams (JP01, KK24, KB91) were intravenously inoculated with 1x10^6 pfu each of RhCMV strains UCD52 and full-length recombinant RhCMV

**Table 1. Study outline of animal groups.**

| | Number of animals | | RhCMV Inoculum | Age in years Mean (Range) | Gestational day Mean (Range) | | |
|---|---|---|---|---|---|---|---|
| | | | | | CD4+ T lymphocyte depletion | RhCMV inoculation | Study termination |
| CMV-seropositive Reinfection | $n$ = 5 | $n$ = 2 | • RhCMV 180.92 | 5.5 [4–9] | 53.2 (49–59) | 60.2 (56–66) | 144 (142–147) |
| | | $n$ = 3 | • RhCMV UCD52 • FL-RhCMVΔRh13.1/ SIVgag | | | | |
| CMV-seropositive Controls | $n$ = 3 | $n$ = 3 | Not inoculated | 7.0 [4–11] | 52.0 (50–55) | Not inoculated | 164 (155–171) |
| CMV-seronegative Primary Infection* | $n$ = 6 | $n$ = 1 | • RhCMV 180.92 | 11.6 [4–16] | 50.5 (47–56) | 57.5 (54–63) | 94 (74–165) |
| | | $n$ = 5 | • UCD52 • UCD59 • RhCMV 180.92 | | | | |

*Bialas, et al. PNAS (2015). Nelson, et al. JCI Insight (2017)

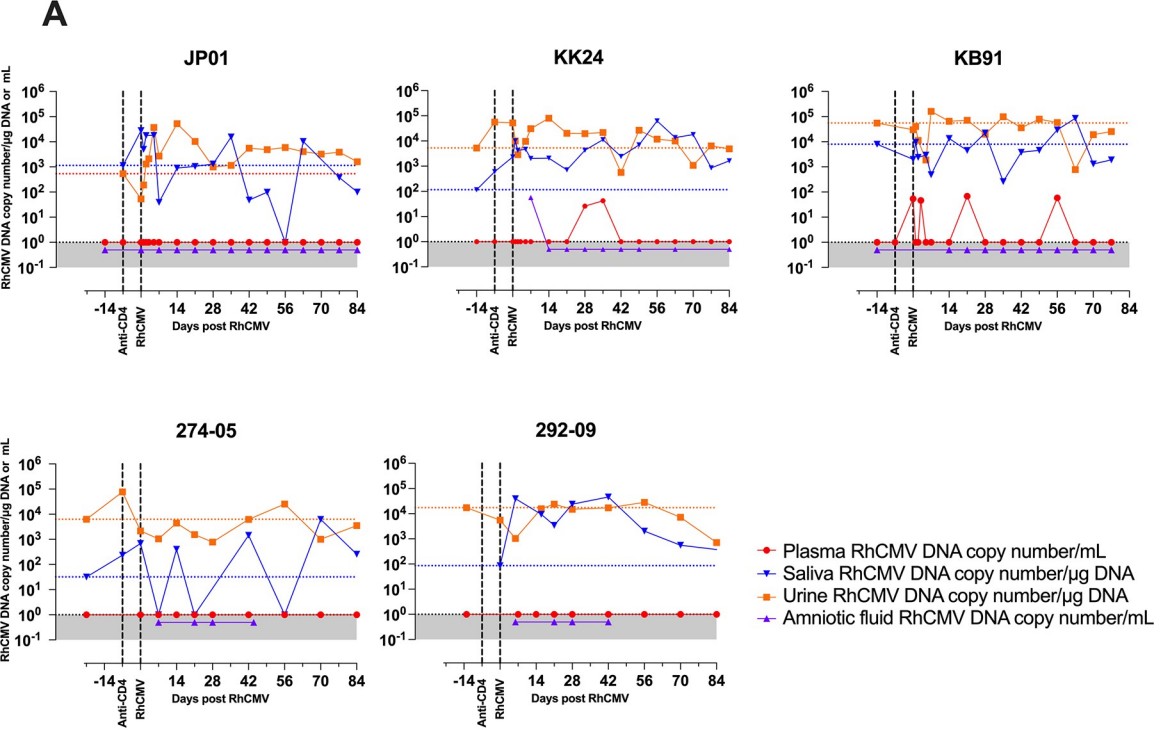

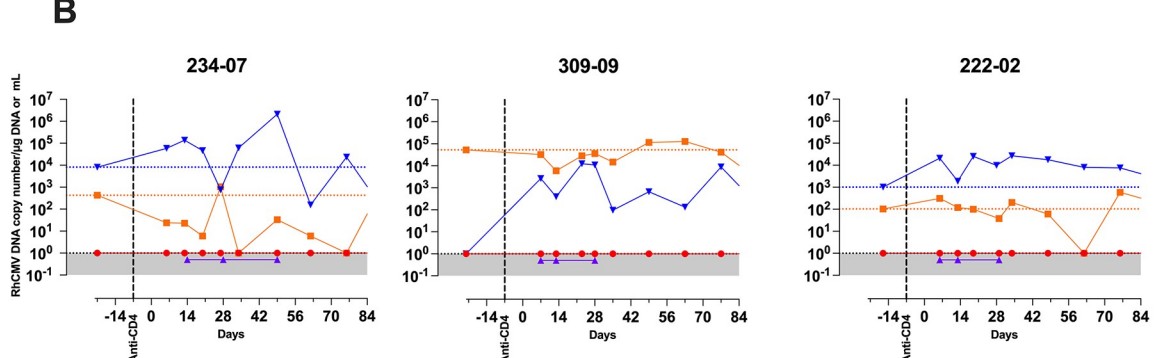

**Fig 2. RhCMV viral kinetics in blood and body fluids of individual CD4+ T lymphocyte depleted CMV-seropositive macaques.**
RhCMV in plasma (indicated in red), saliva (blue), urine (orange), and amniotic fluid (purple) in (A) five CMV-seropositive reinfected animals and (B) three CMV-seropositive control animals. Plasma and amniotic fluid are reported in mean RhCMV DNA copy number/mL of sample fluid; saliva and urine are reported as mean RhCMV DNA copy number/µg of input DNA. In CMV-seropositive controls, the equivalent post-infection time-points on the x-axis are aligned concurrently with the CMV-seropositive Reinfection group. The black vertical lines indicate time of anti-CD4 antibody (CD4R1) and RhCMV inoculation. Animals JP01, KK24, and KB91 were inoculated with RhCMV UCD52 and FL-RhCMVΔRh13.1/SIV*gag*; animals 274–05 and 292–09 were inoculated with RhCMV 180.92; animals 234–07, 309–09, and 222–02 remained without a reinfection. The horizontal stippled line indicates the baseline mean RhCMV DNA copy number/µg of input DNA in either saliva (blue) or urine (orange).

expressing SIV*gag* (FL-RhCMVΔRh13.1/SIV*gag*) administered separately in different limbs, and followed up until an elective Cesarian section (C-section) at mean 144 (range 142–147) days gestation (**Tables 1 and S1, Figs 1A and 2A**). The choice of RhCMV 180.92 and RhCMV UCD52 virus strains were based on previously used isolates shown to cross the placenta in primary infection studies [25, 26, 32]. FL-RhCMVΔRh13.1/SIV*gag*, a full-length RhCMV generated by repairing strain 68–1 with wild-type sequence that includes a heterologous SIV*gag* sequence as a foreign gene in the place of Rh13.1 [27], was used for reinfection experiments in three macaques. The full length clone enabled distinction of endogenous virus from the exogenous challenge inoculum in RhCMV-seropositive animals and allowed confirmation of reinfection as previously demonstrated in RhCMV/SIV vaccine studies [33–35]. Based on the presence of an intact full-length UL/b' region and the *in vivo* replication properties of FL-RhCMVΔRh13.1 in RhCMV-seronegative males being comparable to low passage RhCMV isolates [27], the full-length clone is likely to be the closest approximation to a wild-type human CMV strain that would be encountered during pregnancy. Three RhCMV-seropositive dams (234–07, 309–09, and 222–02) that underwent CD4+ T lymphocyte depletion without reinfection were included as a control group to monitor placental transmission due to reactivation of endogenous RhCMV virus strains (**Tables 1 and S1, Figs 1A and 2B**).

## CD4+ T lymphocyte and RhCMV viral dynamics

Following administration of a single dose of 50mg/kg rhesus recombinant anti-CD4 monoclonal antibody (clone CD4R1), profound CD4 depletion, resulting in loss of >95% of circulating CD4+ T lymphocytes, was observed at day 7 in three out of five RhCMV-seropositive reinfected animals as well as in all three RhCMV-seropositive controls (**Fig 1B and 1C**). The peripheral blood CD4+ T lymphocyte count in these animals declined from a mean baseline value of 950 cells/μL to less than 65 cells/μL throughout the study period. Two RhCMV-seropositive reinfected macaques (JP01 and KK24) showed relatively suboptimal depletion with persistent circulating CD4+ T lymphocytes at 46–68% of baseline values (**Fig 1B**).

RhCMV DNA was monitored in the plasma, saliva, urine and amniotic fluid of reinfection and control dams (**Fig 2A and 2B**). Immunocompetent RhCMV-seropositive macaques are typically aviremic but shed RhCMV in body fluids such as urine and saliva [22]. Following CD4+ T lymphocyte depletion, low levels of plasma RhCMV DNA were detected intermittently in two out of five RhCMV-seropositive reinfection dams (KK24 and KB91) but in none of the three CMV-seropositive control dams (**Fig 2A and 2B**). Placental RhCMV transmission as evidenced by detection of RhCMV DNA in the amniotic fluid was detected in one of five RhCMV-seropositive reinfected dams (KK24) at day 7 post RhCMV inoculation (**Fig 2A**) but not in any of the CD4+ T lymphocyte depletion control animals (**Fig 2B**). Consistent with previous reports, virus shedding as evidenced by detection of RhCMV DNA in the saliva and urine was present in all eight RhCMV-seropositive dams at baseline [22]. A one-log or greater increase in virus shedding in the saliva and / or urine was detected in all five reinfected dams at 1–9 weeks post reinfection (**Fig 2A**). It was also seen in all three control dams post CD4+ T lymphocyte depletion (**Fig 2B**) suggesting that increased viral shedding could result from both reactivation and reinfection. The transmitter dam KK24 had low-level transient viremia post reinfection, however this did not coincide temporally with the time point of detection of RhCMV DNA in the amniotic fluid (**Fig 2B**).

## Innate and adaptive host response to RhCMV reinfection

Several features of innate immune activation were evident within one week of RhCMV reinfection. A 3- to 100-fold non-significant (p-value = 0.25; non-parametric Wilcoxon Signed Rank

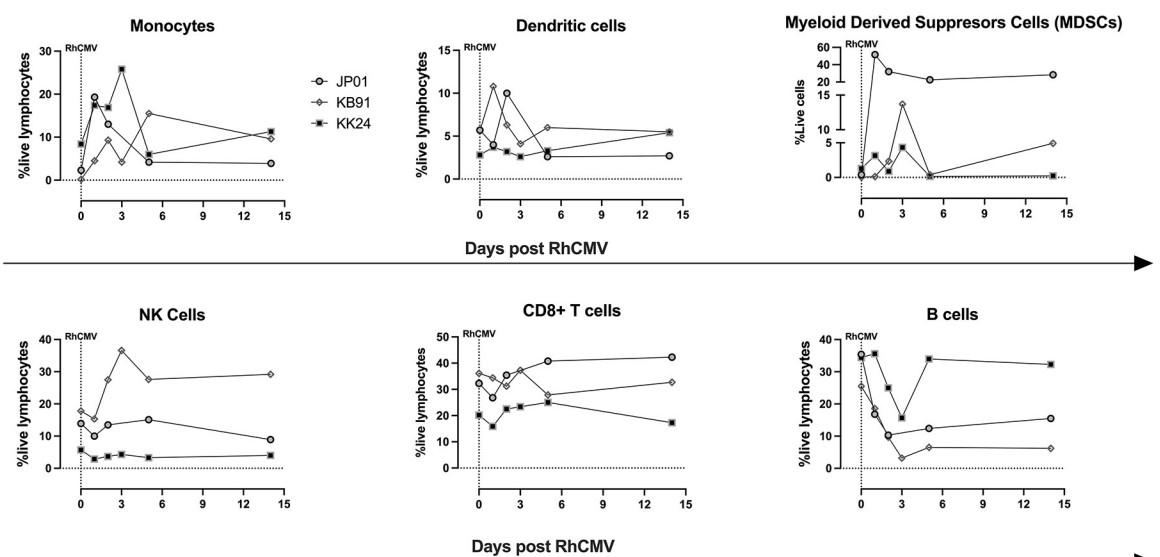

**Fig 3. Early immunophenotypic changes following RhCMV reinfection in CMV-seropositive macaques.** Immunophenotyping of circulating peripheral blood mononuclear cells in acute RhCMV reinfection. Plots show the kinetics of different lymphocyte subsets in three CMV-seropositive reinfected macaques (JP01, KB91, and KK24). Paired non-parametric Wilcoxon Signed Rank test comparing baseline prereinfection values with values at time-point of maximal change in the first 7 days post reinfection was performed.

test) increase in frequency of circulating HLA-DR$^{hi}$CD14$^+$ monocytes from 3.1 ± 3.7% (mean ± SD) to 14.3 ± 10.9% was observed at day 1–5 post reinfection in RhCMV-seropositive pregnant macaques (**Figs 3 and S1A**). Concurrently, there was also an expansion of lineage (Lin)$^-$HLA-DR$^{hi}$ dendritic cells, Lin$^-$HLA-DR$^-$SSC$^{hi}$CD14$^+$ putative myeloid-derived suppressor cells (MDSC) and a transient decline in circulating B lymphocytes which did not reach statistical significance (p-value = 0.25; non-parametric Wilcoxon Signed Rank test) (**Figs 3 and S1A–S1C**). There was also no significant change in the frequency of circulating natural killer (NK) cells and CD8+ T lymphocytes (**Fig 3**). Evidence of innate immune activation was corroborated by an increase in plasma analytes, notably IL-8 and MIP-1b, as measured by Luminex in the first week post reinfection (**S2A Fig**). Of note, an increase in eotaxin was only observed in KK24, the transmitter dam (**S2A Fig**). Overall, these data indicate a rapid activation of the innate immune system predominantly involving myeloid cells in response to RhCMV reinfection.

To evaluate the effect of RhCMV reinfection on pre-existing adaptive immunity, we longitudinally monitored RhCMV-specific CD8+ T lymphocyte responses and anti-RhCMV antibodies in the CD4+ T lymphocyte-depleted reinfection and control macaques. Memory RhCMV-specific CD8+ T lymphocyte responses against the RhCMV immediate early 1 (IE1), IE2 or pp65 peptide pools measured by intracellular cytokine staining assay were detected at baseline in all the CMV-seropositive macaques. RhCMV-specific CD8+ T lymphocyte responses were also measured at 8–10 weeks post reinfection and compared with baseline responses. Four dams in the RhCMV-seropositive reinfection group had baseline immunodominant CD8+ T lymphocyte responses targeting the IE1 or IE2 peptide pools which increased 2- to 10-fold following reinfection indicating a booster effect (**Fig 4**). An increase in CD107a expression, and IFN-γ, IL-2, and TNF-α cytokine secretion was observed post reinfection with a significant increase in the frequency of TNF-α cytokine secreting RhCMV-specific CD8+ memory T lymphocytes (**Fig 4A**). Boolean analysis of all four effector functions showed a trend for decreased proportion of monofunctional and increased proportion of polyfunctional

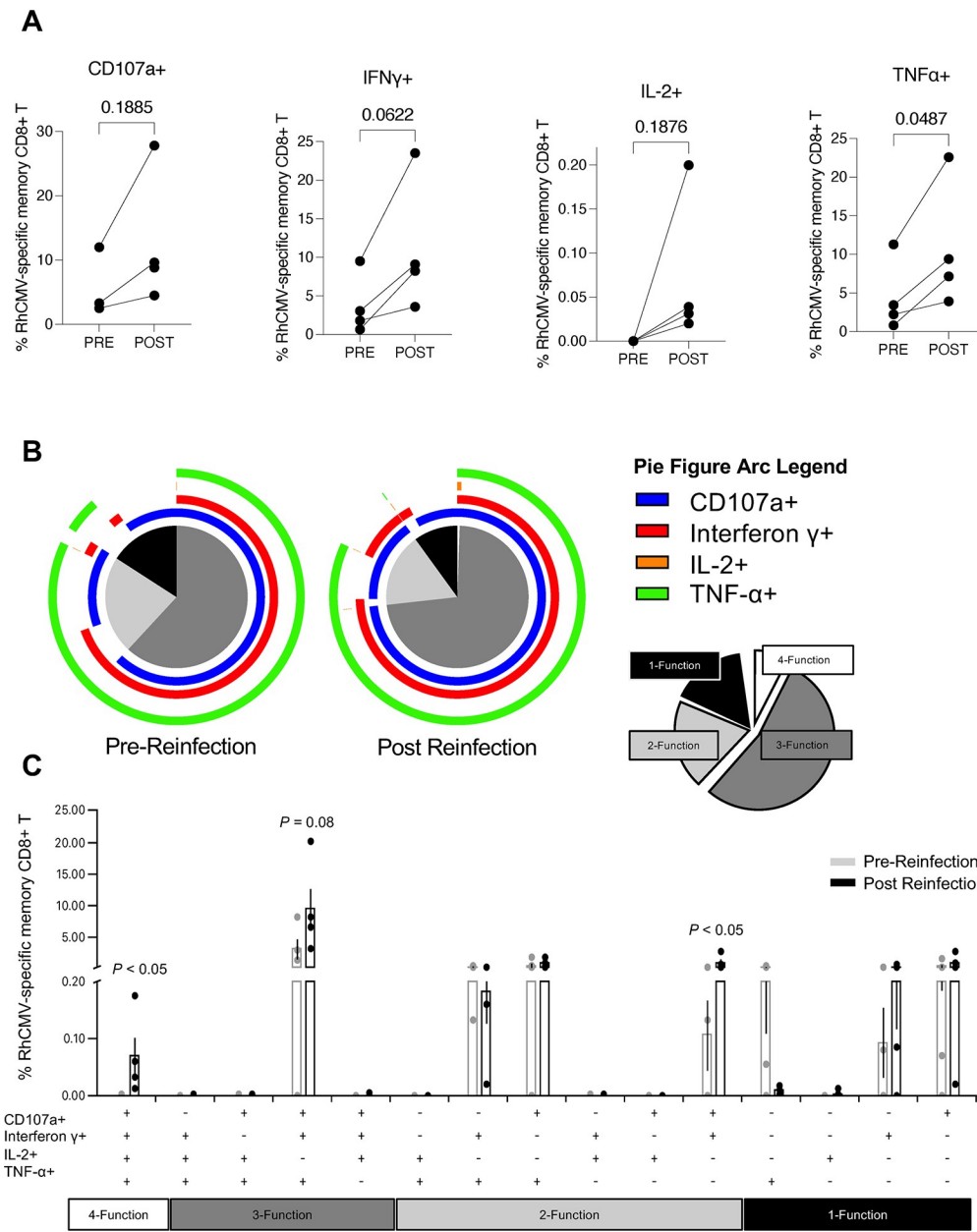

**Fig 4. CMV-specific CD8+ T lymphocyte memory responses to RhCMV immediate early (IE) proteins and exogenous SIV Gag protein in CD4+ T lymphocyte depleted RhCMV reinfected macaques.** (A) Paired IE-specific responses by CD107a expression and secretion of IFN-γ, IL-2, and TNF-α in four CMV-seropositive macaques reinfected with RhCMV UCD52 and FL-RhCMVΔRh13.1/SIV*gag* (*n* = 3) or RhCMV 180.92 (*n* = 1). Pre-reinfection responses were compared with responses at week 8–10 post RhCMV reinfection using paired t-test. (B) Polyfunctional SPICE analysis of IE-specific responses pre vs post RhCMV reinfection showing the proportion of four-functional, three-funtional, two-functional and single function responses. CD107a (blue arc), IFN-γ (red arc), IL-2 (orange arc), and TNF-α (green arc). Four-functional responses are displayed in white, three-functional responses in dark grey, two-functional response in light grey, and mono-functional responses in black. (C) Bar graph of the polyfunctional responses pre (grey) and post (black) RhCMV reinfection (*n* = 4) showing the frequency of memory CD8+ T lymphocytes responding to RhCMV IE peptides. The RhCMV IE-specific response was measured by intracellular cytokine staining after stimulation with RhCMV IE1 and / or IE2 peptide pools depending on the baseline immunodominant response in individual animals. Comparison of pre- and post reinfection Boolean responses were compared with the Wilcoxon rank sum test using SPICE v6 software.

RhCMV-specific CD8+ T cell responses post reinfection (*P*-value 0.11; Fig 4B). This was associated with a significant increase in the frequency of RhCMV-specific 4-functional (*P*-value 0.03) and dual CD107a+IFN-γ+ positive (*P*-value 0.03) CD8+ T lymphocytes post reinfection compared to pre-reinfection values (Fig 4C). Analysis of pre-reinfection RhCMV-specific CD8+ T lymphocyte responses in individual animals revealed that the transmitter dam KK24 had the the lowest frequency (0.91%) as well as the highest proportion of monofunctional responses at baseline compared to the other reinfection dams (S2B Fig). However, the post reinfection boost in this frequency was comparable in the transmitter and non-transmitter dams (S2B Fig). It is noteworthy that the increase in magnitude and polyfunctionality of memory RhCMV-specific CD8+ T lymphocytes post reinfection occurred despite the presence of CD4+ T lymphocyte depletion.

We also monitored end-point dilution titers of anti-RhCMV gB-binding IgG responses and fibroblast neutralization activity against RhCMV 180.92 (Fig 5). Following reinfection, a transient 0.8- to >1.0- log increase in anti-gB binding titers was observed in three RhCMV-reinfected macaques (Fig 5A). Analysis of individual animals showed the two macaques (KK24 and JP01) that experienced a suboptimal CD4+ lymphocyte depletion (46–68% loss) responded with a 1.0-log or greater increase in magnitude of anti-gB IgG responses (Fig 5A), suggesting that CD4+ T lymphocyte help may have aided a boost of anti-gB antibodies against pre-existing and new specifities post RhCMV reinfection. In a group analysis, baseline values for gB-binding and neutralizing antibody levels at pre-CD4+ T lymphocyte depletion time-points were comparable between the RhCMV-seropositive reinfected and RhCMV-seropositive control dams (Fig 5B). Post CD4+ T lymphocyte depletion, a change in antibody titers was only observed in the reinfection dams with significant increases in the first 8 weeks post reinfection compared to the RhCMV-seropositive control dams (Fig 5C and 5D).

In contrast to the reinfected animals, the CD4+ T lymphocyte depleted CMV-seropositive control macaques showed no change in the RhCMV-specific CD8+ memory response post CD4 + T lymphocyte depletion (S3A Fig). Neither did they display an increase in anti-gB IgG levels post CD4+ T lymphpcyte depletion (S3B Fig). In all, our results demonstrate elevation of RhCMV-specific CD8+ T lymphocyte and antibody responses that was evident only after RhCMV reinfection, not reactivation following CD4+ T lymphocyte depletion. These differences could reflect a host response to the high virus inoculum administered to the reinfection dams.

## Placental transmission of reinfection virus

Although the boosting of pre-existing RhCMV-specific immune responses and the increase in viral shedding were suggestive of reinfection, natural fluctuations of endogenous virus replication could also lead to changes in shed virus. To determine if reinfection virus was being shed, we evaluated the three macaques (KB91, KK24, and JP01) that were infected with the clone FL-RhCMVΔRh13.1/SIV*gag* virus containing the exogenous transgene *SIVgag*. Screening the saliva and urine at every time-point in each of the three macaques revealed a low positive signal by SIV*gag*-specific real time PCR in the saliva of one animal, KB91, at a single time-point (**Figs 6A and S4A**). However, SIV Gag-specific IFN-γ-secreting memory CD8+ T lymphocyte responses were detected in all three macaques at 6 weeks or later post reinfection and ranged in frequency between 0.9 to 1.4% of circulating memory CD8+ T lymphocytes (Fig 6B). An antibody assay was also developed to determine if these animals generated antibody responses to the SIV Gag protein. No measureable responses were found in the dams which may be due to a near absence of viremia in the reinfected animals (S4B Fig). Cumulatively, these data provide evidence for successful RhCMV reinfection of the CD4+ T lymphocyte-depleted CMV-seropositive dams.

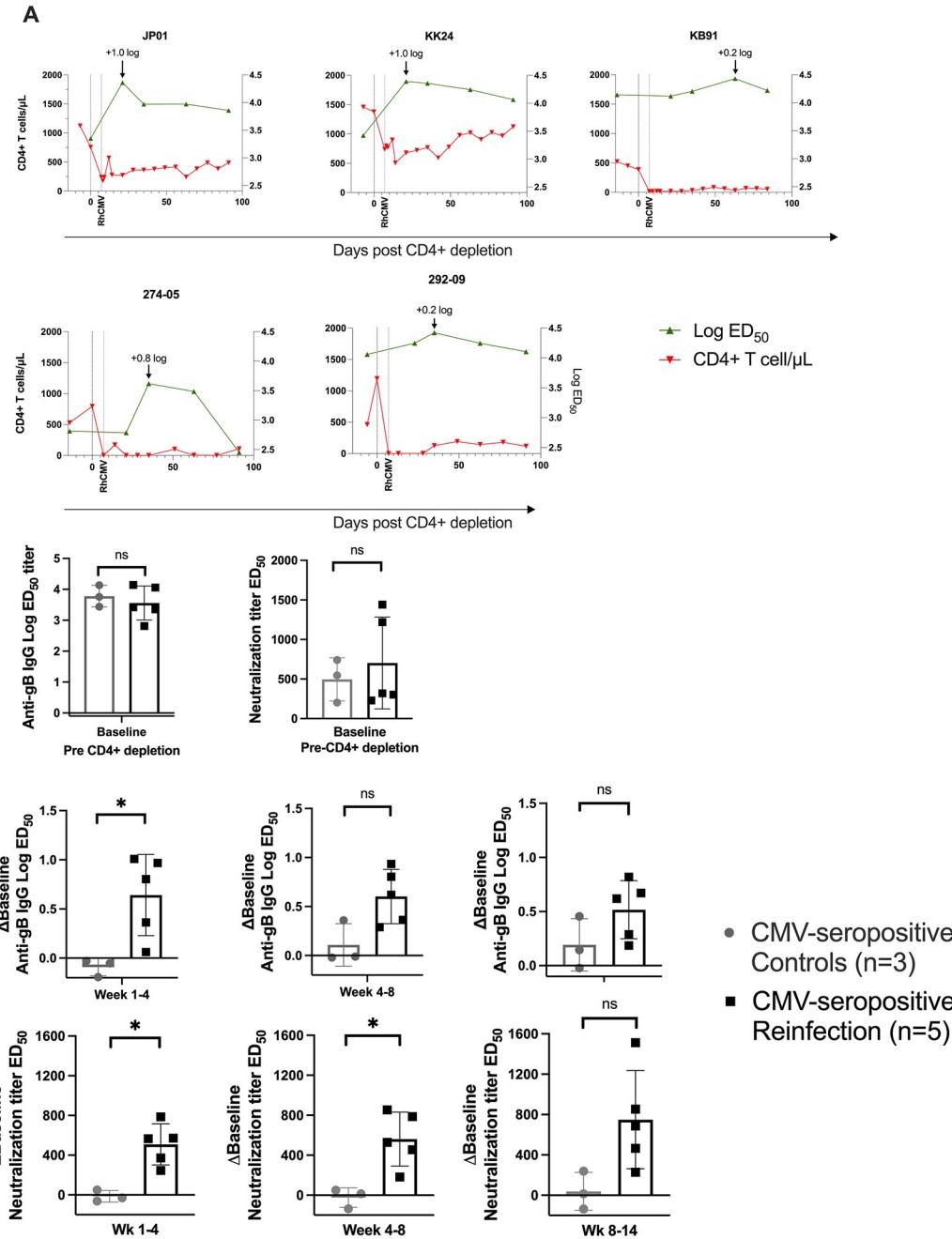

**Fig 5. RhCMV-specific antibody responses in CD4+ T lymphocyte depleted CMV-seropositive reinfected macaques.**
(A) Kinetic data of anti-gB binding antibodies in graphs on individual animals showing log $ED_{50}$ of anti-gB binding titer (green line) superimposed on CD4+ lymphocytes/$\mu$L (red line). (B) Comparison of CMV-seropositive reinfected dams ($n = 5$) and CMV-seropositive controls ($n = 3$) by their anti-gB binding titer and fibroblast neutralization against RhCMV 180.92 at baseline preceding CD4+ lymphocyte depletion. (C) Difference from baseline value in anti-gB IgG $ED_{50}$ titers in the CMV-seropositive reinfection groups at each of the post-infection time-points compared with CMV-seropositive controls at equivalent post CD4-depletion time-points. (D) Difference from baseline value in neutralization titer in the CMV-seropositive reinfection groups at each of the post-infection time-points compared with CMV-seropositive controls at equivalent post CD4-depletion time-points. $ED_{50}$ = Effective Dose 50.

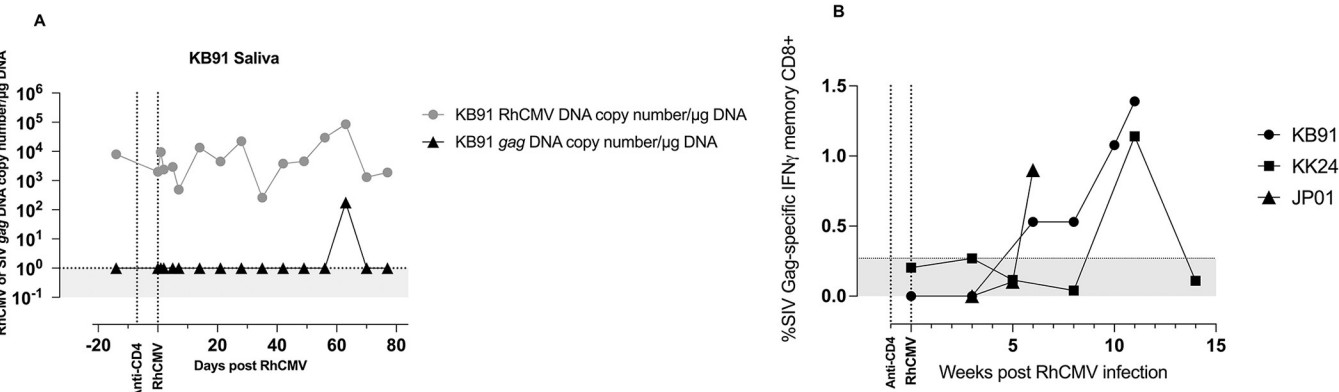

**Fig 6. Evidence of reinfection in CD4+ T lymphocyte depleted FL-RhCMVΔRh13.1/SIV*gag* inoculated dams.** (A) RhCMV-specific (black line) and SIV*gag*-specific (grey line) real time PCR results in the saliva of one CMV-seropositive reinfected animal (KB91). All other animals were found negative for SIV*gag* DNA in saliva and urine. (B) Detection of SIV Gag-specific T lymphocyte responses measured longitudinally against a SIVmac239 Gag peptide pool in CMV-seropositive reinfected macaques (KB91, KK24, and JP01) inoculated with RhCMV UCD52 and FL-RhCMVΔRh13.1/SIV*gag*. Horizontal stippled line shows negative cut-off based on pre-reinfection values.

To confirm the passage of reinfection RhCMV virus across the placenta, amniotic fluid DNA from the transmitter dam KK24 was amplified by multiple displacement amplification (MDA) and viral DNA enriched by PCR-amplification using RhCMV-specific primer pairs. Amplicons were then sequenced to >10,000X coverage on a Ion Torrent Sequencer. The resulting whole genome sequencing data (mean read length: ~200 bp) was mapped against the RhCMV UCD52 and FL-RhCMVΔRh13.1/SIV*gag* reference assemblies. The majority of reads mapped equally well to either reference assembly. However, there are a few uniquely mapping regions that we were able to use to distinguish the two RhCMV strains. Investigations of uniquely mapping regions demonstrated placental transmission of both reinfection RhCMV strains with 68.9% corresponding to RhCMV UCD52 and 31.1% to FL-RhCMVΔRh13.1/SIV-*gag* consistent with the mixed human CMV infections reported in infants with cCMV [36].

## Protective effect of natural pre-existing immunity against congenital CMV transmission

To evaluate protection conferred by pre-conception immunity, we compared viral and pregnancy outcome parameters in the CD4+ T lymphocyte-depleted RhCMV-seropositive reinfection macaques with the CD4+ T lymphocyte-depleted primary RhCMV infection historical control animals (**Fig 7**, **S1** and **S2** **Tables**). The CD4+ T lymphocyte-depleted RhCMV-seropositive reinfected dams showed intermittent, detectable RhCMV DNAemia which was significantly lower compared to RhCMV-seronegative animals with primary infection (**Fig 7A**). The CD4+ T lymphocyte-depleted RhCMV-seropositive controls showed no detectable RhCMV DNAemia (**Fig 7A**). Amniotic fluid sampled at weekly intervals showed variable detection of RhCMV transmission across the study groups as shown in a heatmap (**Fig 7B**). While all six dams in the primary infection group had detectable amniotic fluid RhCMV DNA (mean range 49–580 copies/ml) at one or more sampling time-points, amniotic fluid RhCMV DNA at mean±S.D 57±146 copies/mL was detected in only one animal in the reinfection group at a single time-point (**Fig 7B**). Amniotic fluid RhCMV transmission was not detected in any of the CD4+ T lymphocyte depletion RhCMV-seropositive controls (**Fig 7B**). The fetuses of the mothers with natural pre-existing immunity fared better with 100% fetal survial (**Fig 7C and 7D**). Overall, placental transmission was reduced from 100% in primary infection to 20% in

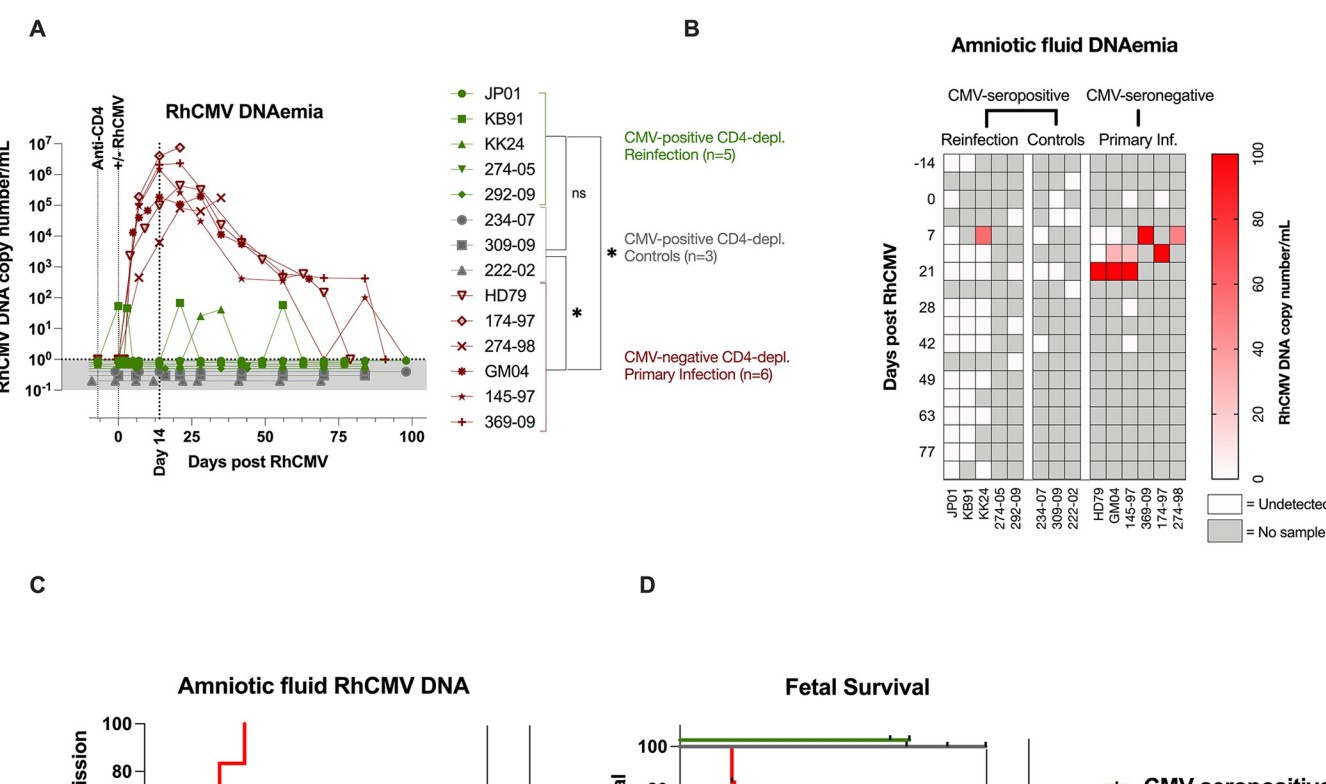

**Fig 7. Protective effect of pre-existing immunity against congenital CMV transmission.** (A) Plasma RhCMV-specific PCR in CD4+ T lymphocyte depleted CMV-seronegative primary infected macaques (red; *n* = 6) compared to both CMV-seropositive reinfected (green; *n* = 5) and CMV-seropositive controls (grey; *n* = 3). Area Under the Curve (AUC) values of plasma RhCMV DNA between 0–99 days were compared between groups using the Man-Whitney test. P-values <0.05 denoted with a single * were considered significant. (B) Heatmap of RhCMV-specific DNA copy number in amniotic fluid in CMV-seronegative primary infected macaques (*n* = 6), CMV-seropositive reinfected (*n* = 5), and CMV-seropositive controls (*n* = 3). (C) Kaplan-Meir curve showing cCMV frequency based on RhCMV DNA detection in the amniotic fluid in CMV-seronegative primary infected macaques (*n* = 6), CMV-seropositive reinfected (*n* = 5), and CMV-seropositive controls (*n* = 3). (D) Kaplan-Meier curve showing fetal survival in CMV-seronegative primary infected macaques (*n* = 6), CMV-seropositive reinfected (*n* = 5), and CMV-seropositive controls (*n* = 3). Statistical comparisons by Log-rank (Mantel-Cox) test showing significance levels: * = <0.05 and ** = <0.01.

the reinfection group (*P* <0.05; Log-rank test), whereas fetal survival was increased from 16% to 100% in the reinfection groups (*P* <0.05; Log-rank test) (**Table 2**, **Fig 7C and 7D**). There was no statistically significant difference in RhCMV DNAemia, placental transmission or fetal

**Table 2. Improved study outcome in dams with pre-existing immunity.** The main readouts of the study are described as a frequency of total number of animals.

| Group name | Natural Immunity to RhCMV | Amniotic fluid PCR RhCMV DNA | Placental and fetal tissue PCR RhCMV DNA | Number of animals |
|---|---|---|---|---|
| **CMV-seropositive Reinfection** | **Yes** | **20%** | **0%** | **5** |
| **CMV-seropositive Controls** | **Yes** | **0%** | **0%** | **3** |
| **CMV-seronegative Primary Infection** | **No** | **100%** | **100%** | **6** |

survival between the RhCMV-seropositive controls and reinfection group. The small group sizes precludes definitive conclusion regarding the contribution of reinfection *vs* reactivation to non-primary cCMV infection. RhCMV DNA PCR evaluation of placental tissues and fetal tissues confirmed a protective role of maternal pre-existing immunity in preventing the spread and replication of RhCMV *in vivo* during first trimester infection of rhesus macaques (**S2 Table**). The fetal growth parameters in the CD4+ T lymphocyte-depleted RhCMV reinfected dams were comparable to the RhCMV-seropositive control dams (**S5 Fig**) and within the normal range of reference values established in rhesus macaques [37].

## Discussion

In this study we provide the first evidence of placental transmission in a nonhuman primate model of non-primary cCMV and demonstrate that pre-conception maternal CMV-specific immunity from previous natural CMV infection has a protective effect against congenital transmission and disease. By conducting a longitudinal experimental superinfection study with an inoculum of defined RhCMV virus stocks administered at a known gestation time-point in pregnant RhCMV-seropositive rhesus macaque dams, we could determine the risk of vertical transmission after non-primary CMV infection and compare it with historical controls from our previous studies of primary cCMV in this animal model. Moreover, by including a RhCMV virus with a foreign transgene (SIV*gag*), we could track one of the inoculated RhCMV viruses *in vivo* and distinguish it from endogenous RhCMV in the reinfected animals. Our data show that even in the setting of CD4+ T lymphocyte depletion, RhCMV-reinfected RhCMV-seropositive dams showed only 20% placental transmission and did not suffer any adverse pregnancy outcome or fetal infection. These findings are in sharp contrast to primary RhCMV infection in CD4+ T lymphocyte-depleted CMV-seronegative rhesus macaque dams, which resulted in 100% placental transmission and 83% fetal loss [25, 26]. In the absence of CD4+ T lymphocytes, the protective effect of prior natural RhCMV infection was accompanied by an initial activation of the innate immune system in the first week after reinfection followed by boosting of pre-existing memory RhCMV-specific CD8+ T lymphocyte and antibody responses. These data point to redundant pathways of immune-mediated protection and suggest that vaccine approaches harnessing different arms of the immune system will be needed to prevent cCMV infection.

RhCMV-seropositive pregnant macaques model the immunity of reproductive age women in a high CMV seroprevalence settings. The RhCMV-seropositive macaques used in this study acquired naturally circulating RhCMV strains prevalent in the primate center colony and were never experimentally infected prior to this study. Thus, their pre-existing immune status was a result of natural RhCMV infection akin to human populations susceptible to non-primary infections. As demonstrated previously, reinfection of RhCMV-seropositive animals is enabled by viral T cell evasion mechanisms [35]. However, although T cells cannot prevent reinfection due to viral immune evasion there was a clear impact of pre-existing immunity on congenital infection. In the current study, we saw a 5-fold reduction in transmission rate after non-primary infection as compared to primary infection despite the presence of CD4+ T lymphocyte depletion–akin to a recent study in humans [12]. However, unlike in humans, we did not find any evidence of fetal infection or pathology. Because the diagnosis of non-primary cCMV in humans is based on the screening of newborns, and because there are no data on amniotic fluid CMV load monitoring in pregnant CMV-seropositive women, we cannot exclude the possibility of placental transmission without fetal infection also occurring in human subjects. The reduction in transmission and absence of fetal infection is indicative of a protective role of natural pre-conceptional immunity. However, there are caveats to extrapolating the findings

in the NHP model to non-primary cCMV in humans. The small number of animals in this study precludes definitive conclusion of the transmission rate following non-primary infection. Furthermore, the risk of placental transmission following a single reinfection event, as was the design of the current study, does not capture the potential of multiple reinfections occurring in CMV-seropositive pregnant women in high CMV seroprevalence regions when exposed to frequent high levels of CMV shedding from different sources. Annualized CMV seroconversion rates in CMV-seronegative women are directly related to the extent of exposure to CMV shedding with 20% or higher rates in high CMV seroprevalence areas [14, 38]. The same pattern may hold true for reinfection rates.

The extent to which non-primary CMV infection is a result of reactivation or reinfection, or a combination of both factors, is not known. In one of the first studies to document reinfection as an important cause of non-primary cCMV, 62% of mothers with cCMV births compared to 13% with normal births developed new antibody specificities against glycoprotein H during pregnancy suggestive of reinfection [4]. More recently, the contribution of reinfection to non-primary cCMV has been deduced from detection of new antibody specificities to strain-specific polymorphic regions in the glycoproteins gB and gH [5, 6]. Using these criteria, the annualized rate of reinfection in CMV-seropositive women has ranged between 10–35% in different studies with reinfections being associated with an increased risk of cCMV [5, 6]. In addition to reinfection, new strain-specificities could also be due to emergence of a previously subdominant or quiescent endogenous CMV due to reactivation, but this has not been established. Our model looked at both modes of non-primary CMV infection in pregnant rhesus macaques, albeit in a small number of animals. As controls for the group of CD4+ T lymphocyte-depleted reinfected macaques, we used CD4+ T lymphocyte depletion alone to model endogenous CMV reactivation without reinfection. We inferred reactivation from a one-log or greater increase in RhCMV load shed in the urine or saliva at one or more post CD4+ T lymphocyte depletion time points. Unlike adult humans where CMV shedding is detected only in a subset of individuals, group-housed RhCMV-seropositive rhesus macaques invariably have detectable RhCMV DNA shed in the urine and/or saliva [22, 39]. Consistent with this observation, all the RhCMV-seropositive macaques enrolled in this study had detectable presence of RhCMV DNA shed in the urine and/or saliva at baseline prior to any intervention. A one-log or greater increase in RhCMV shedding at post CD4+ T lymphocyte depletion time-points was detected in both the reinfection and control group of animals suggestive of reactivation. Although transient viremia and amniotic fluid transmission were only detected in the RhCMV-seropositive, CD4+ T lymphocyte depleted, reinfected dams, the results were not statistically different from the RhCMV-seropositive, CD4+ T lymphocyte depleted controls. One of the viremic reinfected dams, macaque KK24, transmitted the virus across the placenta. Confirmation of reinfection was supported by the emergence of immune responses against the foreign transgene present in the inoculated RhCMV virus, and detection of this foreign gene in shed virus. These findings were confirmed by detection of the reinfection virus in the amniotic fluid of the transmitter dam by genome-wide deep sequencing. In the absence of sequencing RhCMV shed in the urine and saliva prior to reinfection we cannot ascertain if endogenous reactivated RhCMV also crossed the placenta in the transmitter dam. Despite detection of placental transmission after reinfection, the small group size precludes definitive conclusions about the relative contribution of reinfection or reactivation to non-primary cCMV. Moreover, the high dose of viral inoculum administered intravenously could have contributed to detection of placental transmission in the reinfection animals as opposed to the CD4+ T lymphocyte-depleted RhCMV-seropositive controls.

Sequencing analysis confirmed that the two RhCMV strains used for reinfection had both crossed the placenta, with RhCMV UCD52 being more prevalent than FL-RhCMVΔRh13.1/

SIV*gag*. Passage of the RhCMV UCD52 strain was consistent with the findings in our primary infection studies where UCD52 was the dominant virus in the circulation and amniotic fluid [25, 26]. This raises the question of what might determine placental transmission of reinfection virus. Aside from protective immunity, viral diversity and inoculum size are likely to be important determinants of cCMV transmission after reinfection. Recent studies have shown the surprising diversity of naturally circulating HCMV, which can be in the range of that observed in RNA viruses such as dengue virus and HIV [40]. Moreover, mixed strain infections, such as the one observed here, are common in CMV-seroimmune individuals and reinfection cCMV [36, 41, 42]. The presence of significant diversity between RhCMV strains may be an important factor determining which virus strains are transmitted. For example, viral diversity of the reinfection RhCMV from the endogenous circulating strains could render it less susceptible to immune control and hence more transmissible. Based on the sequencing data, the bottleneck of transmission appears to be wide as both virus inoculums were detected in the amniotic fluid. This phenomenon is similar to our findings in primary infection in CD4+ T lymphocyte-depleted dams where the genetic composition of RhCMV in the blood and amniotic fluid after a mixed infection with three RhCMV viruses was similar and did not reveal a readily detectable transmission bottleneck even in the presence of immune pressure exerted by pre-existing passively infused antibodies [32]. However, with the current data we are unable to exclude the endogenous virus from being present in the transmitting sequences. Future sequencing analysis of the endogenous vs challenge sequences are required to answer this and to determine the relative divergence between endogenous and inoculum viruses between the animals.

Despite the absence of CD4+ T lymphocytes, RhCMV-seropositive macaques showed evidence of protection against cCMV transmission and infection. The protective effect was likely mediated by both innate immunity and CMV-specific adaptive immunity. Protection in the absence of CD4+ T lymphocytes points to protection mediated by the CD8+ T lymphocyte and humoral components of adaptive immunity. It is noteworthy that despite CD4+ T lymphocyte depletion and potential lack of CD4 help, there was still a robust memory CMV-specific CD8+ T lymphocyte or the antibody response post reinfection. Boosting of CD8+ T lymphocyte and antibody responses were only seen in reinfection animals and not in the CD4+ T lymphocyte depletion control animals. The endogenous anti-gB IgG antibodies and the pre- reinfection CMV-specific CD8+ T lymphocyte responses were significantly increased following reinfection when compared to controls. The increase in gB-specific IgG and RhCMV-specific CD8+ T lymphocyte responses is unlikely to have been a result of RhCMV reactivation as it was not observed in the CD4+ T lymphocyte depleted controls without reinfection. Of note, rhesus macaques that were both CD4- and CD8 T lymphocyte-depleted preceding kidney transplant did not see a rise in anti-gB IgG titer, suggesting that it is unlikely that our observation of boosted immune responses are a consequence of reactivation [43]. It is interesting that the magnitude of the increase in antibody responses was greatest in the two macaques with partial CD4+ T lymphocyte depletion suggesting a facilitatory role for CD4 help in potentiating humoral immunity. The magnitude of the boosted memory CMV-specific CD8+ T lymphocyte response however did not appear to be affected by the extent of CD4+ T lymphocyte depletion. In the absence of epitope mapping, we cannot determine whether the amplified responses were anamnestic in nature with expansion of pre-existing or cross-reactive specificities against reinfection virus or included new responses directed towards new epitopes in the reinfecting strain.

The small number of animals in this study with only one placental-transmitter dam prevents analysis of protective correlates based on differences in immunity between transmitters and non-transmitters. Observationally, the placental-transmitter dam (KK24) displayed

certain features that were different from the non-transmitter dams. For example, an increase in plasma eotaxin following reinfection was only detected in the transmitter dam. KK24 also had the lowest frequency of RhCMV IE-specific CD8+ T lymphocyte responses at baseline prior to reinfection. In addition, KK24 was one out of two dams that showed transient viremia post reinfection and that had a partial CD4+ T lymphocyte depletion as well as a 1.0 log or greater increase in anti-gB IgG response following reinfection. Whether this could have resulted in an increase in low affinity binding antibodies and transmission facilitated by transcytosis of immune complexes is an interesting possibility. Studies with a larger group of animals are warranted.

In conclusion, establishment of the NHP reinfection cCMV model has provided definitive evidence for a role of pre-conceptional natural immunity in CMV-seropositive individuals to partially protect against cCMV. Strikingly, this protection is evident even in the absence of CD4+ T lymphocytes and likely involves multiple arms of the immune system including CD8 + T lymphocyte-mediated and humoral immunity. Importantly, the establishment of this model lays the groundwork for future experiments including CD8+ T lymphocyte depletion and B lymphocyte depletion studies in CMV-seropositive macaques to dissect the contribution of different arms of the adaptive immune system involved in protection against non-primary cCMV. Taken together, our data reinforces the utility of the rhesus macaque model in furthering knowledge about immune determinants of cCMV protection, needed for rational vaccine design.

## Methods

### Ethics statement for *in vivo* non-human primate studies

*In vivo* non-human primate studies were performed at the New England Primate Research Center (NEPRC) and the Tulane National Primate Research Center (TNPRC). This study was carried out in accordance with the Guide for the Care and Use of Laboratory Animals of the NIH. The animal protocol was reviewed and approved by the Institutional Animal Care and Use Committees (IACUCs) at NEPRC and TNPRC. The facilities also maintained an Animal Welfare Assurance statement with the National Institutes of Health, Office of Laboratory Animal Welfare.

### Animals and study design

A total of eight CMV-seropositive Indian-origin first trimester of gestation rhesus macaques were enrolled in the study from the specific pathogen free colony at TNPRC and NEPRC (Table 1). Gestational age at enrollment was estimated by ultrasound measurement of gestational sac diameter and/or crown-rump length. Subsequently, the gestational age was monitored weekly by ultrasound measurement of biparietal diameter and femur length. At Caesarian section (C-section), measurements of occipitofrontal diameter, head circumference, and abdominal circumference were recorded.

All eight dams were subjected to *in vivo* CD4+ T lymphocyte depletion by intravenous (IV) administration of 50 mg/kg rhesus-recombinant anti-CD4 antibody (Clone CD4R1; Nonhuman Primate Reagent Resource) at mean 52.3 gestation days (range 49–59). The rhesus IgG1 recombinant Anti-CD4 [CD4R1] monoclonal antibody was engineered and produced by the Nonhuman Primate Reagent Resource (NIH Nonhuman Primate Reagent Resource Cat# PR-0407, RRID:AB_2716322).

Five of eight dams were reinfected with RhCMV one week after administration of the CD4-depleting antibody (CMV-seropositive reinfection group), whereas the remaining three dams that received the CD4-depleting antibody were not reinfected and served as a control

group for reactivation following CD4+ T lymphocyte depletion (CMV-seropositive Controls). Data from six dams with CD4-depletion and a primary infection with RhCMV served as historical controls (**Table 1**) [25, 26].

One week after administration of the anti-CD4 antibody, the CMV-seropositive reinfection group were inoculated IV with $2x10^6$ TCID$_{50}$ of RhCMV 180.92 (*n* = 2) or $1x10^6$ pfu each of RhCMV UCD52 and FL-RhCMVΔRh13.1/SIV*gag* (*n* = 3; **Table 1**). Maternal blood, saliva, urine, amniotic fluid were collected preceding anti-CD4+ T lymphocyte depletion and virus inoculation. Following RhCMV inoculation the animals were sampled weekly for all sample types. Three animals that received the RhCMV UCD52 and FL-RhCMVΔRh13.1/SIV*gag* were in addition sampled for blood, urine and saliva every 1–3 days within the first 7 days post infection to study the acute phase of infection. All animals were subsequently followed with weekly sampling until term or C-section, outlined in **Table 1**.

## Virus stocks for reinfection

Full Length-RhCMV68-1-ΔRh13.1/SIV*gag* (FL-RhCMVΔRh13.1/SIV*gag*) virus was generated and characterized as previously described [27] and administered intravenously at a single dose of $1x10^6$ pfu. RhCMV strain UCD52 virus stock grown in TeloRF cells was provided by Dr. Peter Barry [31] and administered intravenously at a single dose of $1x10^6$ pfu. Each virus strain was inoculated in to separate limbs of three animals (**S1 Table**). RhCMV 180.92 virus stock grown in primary rhesus fibroblasts as previously described [28, 29] was administered at a single dose of $2x10^6$ TCID$_{50}$ intravenously to two animals.

## Sample collection and processing

Maternal PBMC were isolated by ficoll separation after collecting plasma. All PBMC were cryopreserved using 90%FBS/10% DMSO. Amniotic fluid was spun to remove debris prior to storage in aliquots at -20˚C. Saliva and urine sample supernatants were concentrated using Ultracel YM-30 (Amicon/Milipore) and subsequently aliquoted for storage at -20˚C for DNA extraction. At C-section, the placenta and fetal tissues were harvested and processed for lymphocyte isolation, snap frozen for DNA extraction for PCR and placed in Z-fix for paraffin blocks.

DNA was extracted from urine with the QIAmp RNA mini kit (Qiagen, Velencia, CA); from amniotic fluid, plasma, and saliva with the QIAmp DNA mini kit (Qiagen, Velencia, CA); and from 10-25mgm of snap-frozen tissue using the DNeasy Blood and Tissue kit (Qiagen) as previously described [22, 25].

## Viral quantitation by real time PCR

Absolute quantification of RhCMV DNA in tissues and maternal fluids were performed as previously described [25, 44]. Briefly, for RhCMV DNA PCR, the primers/probe targeting the noncoding exon 1 region of the immediate early gene were used. The Forward primer `5′-GTTTAGGGAACCGCCATTCTG-3′`, Reverse primer `5′-GTATCCGCGTTCCAATGCA-3′`, and probe `5′-FAM-TCCAGCCTCCATAGCCGGGAAGG-TAMRA-3′` were used in a 25μL reaction with Supermix Platinum Quantitative PCR SuperMix-UDG (Invitrogen). The reaction was performed in a 96-well format for real time quantification on Applied Biosystems 7900HT Fast Real-Time PCR System. A standard curve generated from amplification of $10^5$–$10^°$ copies of plasmid standard containing the RhCMV IE target sequence diluted in genomic DNA from CMV-seronegative rhesus macaques was used for absolute quantitation of RhCMV DNA. The lower limit of detection of the PCR assay is between 1–10 copies of RhCMV DNA in the PCR reaction. Real time PCR was performed in 6–12 replicates and at

least 2 positive replicates were required to be reported as a positive result. Data of plasma and amniotic fluid were reported as mean RhCMV DNA copies per mL of sample while saliva, urine, and tissues were reported as mean RhCMV DNA copies per microg of input DNA.

For absolute quantification of SIV*gag* DNA in the three animals which received FL-RhCMVΔRh13.1/SIV*gag* both a nested and real-time PCR protocol was performed. A plasmid carrying SIV*gag* sequence was synthesized (Intregrated DNA Technology, Iowa) and used as standard in a 25uL reaction with Platinum Taq DNA polymerase (cat# 10966034 Invitrogen) mastermix containing 0.012% Tween 20, 0.006% gelatin, 4.5mM $MgCl_2$, 300μM dNTPs, 10% PCR II buffer, 300nM Forward 5′-CAACTACGTCAACCTGCCACTGTC-3′, 300nM Reverse 5′-TCCAACGCAGTTCAGCATCTGG-3′, 200nM Probe 5′-FAM-CCGAGAACC CTGAACGCTTGGGTCAAGC-3BHQ-3′. This was performed in 96 well plate format on Applied Biosystems 7900HT Fast Real-Time PCR System at 95˚C for 2 minutes and cycled for 45 cycles at 95˚C for 15 seconds, and 60˚C for 1 minute.

## Viral sequencing and analysis

Ultradeep sequencing of RhCMV DNA in the amniotic fluid of KK24 generally followed amplicon based methodologies previously established for sequencing human CMV genomes [45] as applied to RhCMV [46]. These studies demonstrated that the error rates observed in amplicon based sequencing and direct sequencing of BAC clones for both viral species are very low and produce nearly identical results. Thus, errors resulting from the amplicon-based workflow plus sequencing are very similar in type and rate to those introduced by direct sequencing and applicable to differentiating viral strains in a sample. As the amount of viral DNA in the KK24 amniotic fluid was limited, MDA (Repli-g, Qiagen) was initially used to increase the overall amounts of DNA for the sequencing workflow. The resulting DNA was repurified and subjected to plate PCR using primer pairs that span the parental RhCMV genome and the SIV*gag* insert, fragmented and then ligated to Ion Express barcodes (Life Technologies). Products were pooled with final processing performed on an Ion Chef (Life Technologies) followed by sequencing on an Ion Proton Sequencer (Life Technologies).

Whole-genome sequencing data was mapped against the RhCMV UCD52 and FL-RhCMVΔRh13.1/SIV*gag* reference assemblies using NextGenMap v.0.5.15 [47]. Thereby, the human (hg38) and rhesus (rheMac10) genomes (downloaded from NCBI GenBank using accession numbers GCA_000001405.29 and GCA_003339765.3, respectively) were included as decoys to remove any potential contamination. To distinguish between the RhCMV strains, uniquely mapping regions were identified using SAMtools v.1.12.0 [48].

## Cytokine and Chemokine analysis Luminex

To analyze peripheral soluble cytokines and chemokines, a luminex assay was performed with Cytokine & Chemokine 30-Plex NHP ProcartaPlex Panel (Invitrogen, EPX300-40044-901). The analytes in this panel are BLC (CXCL13); Eotaxin (CCL11); G-CSF (CSF-3); GM-CSF; IFN alpha; IFN gamma; IL-1 beta; IL-10; IL-12p70; IL-13; IL-15; IL-17A (CTLA-8); IL-18; IL-1RA; IL-2; IL-23; IL-4; IL-5; IL-6; IL-7; IL-8 (CXCL8); IP-10 (CXCL10); I-TAC (CXCL11); MCP-1 (CCL2); MIG (CXCL9); MIP-1 alpha (CCL3); MIP-1 beta (CCL4); sCD40L; SDF-1 alpha (CXCL12a); TNF alpha. Plasma was thawed on ice and manufacturers instructions were followed to prepare a 96-well plate with samples performed in duplicates and read on a Bio-Plex 200 System (Bio-Rad Laboratories, Hercules, CA). Results were calculated using Bio-Plex Manager Software v6.2 (Bio-Rad) and the mean concentration of each analyte was plotted.

## Immunophenotyping and intracellular cytokine staining (ICS) assays

CD4+ T lymphocyte depletion kinetics were monitored by flow cytometric evaluation of absolute counts. Briefly, 50 μL whole blood was stained with an 8-color panel of FITC-CD3, PerCP-CD45, APC-CD4, V500-CD8, PE-Cy7-CD95, APC-CY7-CD20, BV421-CCR7. A FMO was performed for CCR7 which was first stained alone for 15 minutes and then with the remaining cocktail for an additional 15 minutes. Red blood cells were lysed using BD Lysing buffer for 15–20 minutes and subsequently aquired on a BD FACSverse.

A 13-color flow cytometry panel was used for immunophenotyping of the acute reponses following RhCMV infection. PBMCs were stained with the following antibodies: FITC-Ki67, PCP-Cy5.5-TCRgd, APC-KIR2D, AL700-Granzyme B, APC-CY7-CD3, PacBlue-CD20, BV510-live/dead, BV605-CD14, BV650-CD8, BV711-CD16, PE-CD169, PE-CF594-HLA-DR, PE-CY7-NKG2A (for additional details regarding the antibodies, see **S3 Table**). These data were aquired on the BD LSRFortessa and analyzed using Flowjo v9.9 (Ashland, Oregon).

Antigen-specific T lymphocyte respones were assesed by intracellular cytokine staining for RhCMV-specific and SIV Gag-specific responses. Cryopreseved PBMC were thawed and stimulated for 12–18 hours with RhCMV IE1, IE2, pp65 and SIV Gag peptide pools. Briefly, the RhCMV peptide pool and the SIV Gag peptide pool consists of pools of 15-amino acid long peptides, overlapping by 11 amnio acids and spanning the entire protein. Peptides were used at a concentration of 1μg/mL of individual peptides for stimulation and DMSO concentrations were kept <0.5%. After 1 hour, monensin 2μM/mL (cat# 554724, BD) and brefeldin A 1μL/mL (cat#555029, BD) were added along with CD107a-FITC and CD107b-FITC for the remaining period of stimulation. After stimulation, cells were washed and stained sequentially with the following: AQUA Live/dead dye, surface, and intracellular cytokine staining antibodies PCP-Cy5.5-CD4, APC-CD69, AL700-TNFα, APC-CY7-CD3, BV421-Granzyme B, BV605-IL-2, BV650-CD8, BV711-CD95, PE-CCR5, PE-CF594-CD28, PE-CY7-IFNγ using the BD fix/perm kit(BD Cat# 554714) and Brilliant Stain buffer(BD Cat# 563794). These data were acquired on the BD LSRFortessa and analyzed using Flowjo v9.9 (Ashland, Oregon). Boolean analyses of polyfunctional responses were performed using the SPICE 6 software [49].

## Antibody assays for RhCMV gB and SIV Gag

**RhCMV gB.** DNA plasmid expressing RhCMV gB was transfected into 293F/293i cells. Protein was purified with nickel beads and quantitated on a NanoDrop.

**IgG ELISA.** ELISAs were performed as previously described [25]. Briefly 384-well high protein binding plates (Corning 3700) were coated overnight at 4˚C with either 30 ng RhCMV gB produced as described above or SIVmac251 pr55 Gag recombinant protein (NIH AIDS Reagent Program cat# 13384). Serially diluted plasma samples were added to the coated ELISA plates, incubated for 2 hours after which the plates were washed, and then incubated with the secondary anti-monkey IgG HRP-conjugated antibody followed by addition of substrate as previously described [25]. ELISA titers were reported as the serum dilution yielding 50% maximum absorbance (Effective Dose $ED_{50}$). $ED_{50}$ was determined using non-linear regression in GraphPad Prism v9.5. Data are reported as $log^{10}ED_{50}$.

**Antibody neutralization assay.** Telo-RF cells were seeded in a 384-well plate and incubated for 1 day at 37˚C at 5% CO2. The next day, serial dilutions (1:10–1:21,870) of heat-inactivated rhesus plasma were incubated with 1 PFU of RhCMV 180.92 per cell. Infected cells were then fixed for 20 minutes at -20˚C with 1:1 methanol/acetone, rehydrated in PBS three times for 5 minutes and stained with 1 mg/mL mouse anti-RhCMV-IE1 monoclonal antibody provided by Dr. Klaus Früh (Oregon Health and Science University, Portland, OR) followed by a 1:1000 dilution of anti-mouse IgG-Alexa Fluor 488 antibody. Nuclei were stained with DAPI

for 5 minutes (Pierce) and imaged using the CellInsight CX5 High-Content Screening (HCS) platform. The 50% neutralization titer (NT50) was determined by comparing the dilution that resulted in a 50% reduction in fluorescence signal to control wells infected with virus only.

## Statistics

Unpaired and paired parametric and non-parametric t-tests were performed in GraphPad Prism version 8.4.0. (San Diego, CA). P-values <0.05 were considered significant. For viral load the area under the curve (AUC) was calculated between day 0 and 99 days post infection for inter-group comparison. Kaplan-Meier survival curves were calculated in Graphpad Prism for transmission events and fetal survival and statistical comparisons performed with the Log-rank (Mantel-Cox) test.

## Supporting information

**S1 Fig. Gating strategy for PBMC immunophenotyping analysis in acute RhCMV reinfection.** (A) Gating strategy used for the innate immune cell compartment. (B) The gating strategy used to identify T cell subsets. (C) Representative plots of side scatter high (SSChi) population following RhCMV reinfection in CMV-seropositive rhesus macaque dams.
(TIF)

**S2 Fig. Innate and adaptive immune responses in CMV-seropositive reinfected dams.** (A) Plasma IL-8, MIP-1b, Eotaxin, and B-lymphocyte chemoattractant (BLC) levels in three RhCMV reinfected dams in the first two weeks post RhCMV reinfection. Data generated using a nonhuman primate (NHP) 30-plex Luminex assay. Limit of detection (LOD) of the lot# is shown as a stippled line at the bottom of the y-axis. (B) Total memory RhCMV IE-specific CD8+ T lymphocyte responses at pre- and post reinfection time-points in four dams shown in the top panel. Post reinfection time-points varied between week 8 (KK24, KB91, JP01) and week 10 (274–05) post reinfection. Bottom panel showing pie charts depicting proportion of 3-functional, 2-functional and mono-functional RhCMV IE-specific CD8+ T lymphocyte responses prior to reinfection in the four dams.
(TIF)

**S3 Fig. RhCMV-specific adaptive immune responses in CD4+ T lymphocyte depleted CMV-seropositive control macaques.** (A) Memory CD8+ T lymphocyte responses to RhCMV IE protein in CMV-seropositive controls. (B) Kinetics of RhCMV gB-specific binding antibodies in CMV-seropositive controls.
(TIF)

**S4 Fig. Gag-specific PCR and binding antibody assay in CD4+ T lymphocyte depleted CMV-seropositive rhesus macaque dams that received RhCMV FL-RhCMVΔRh13.1/SIV-gag.** (A) Real time PCR for SIVgag DNA quantification performed on plasma, saliva and urine at multiple post reinfection time-points in three dams inoculated with FL-RhCMVΔRh13.1/SIVgag. Positive signal at a single time-point in KB91 saliva sample. (B) Gag-specific binding antibody assays in the three dams reinfected with FL-RhCMVΔRh13.1/SIVgag. Positive controls in this assay are SHIV-infected rhesus macaques (open symbol) were used for comparison of responses in CMV-seropositive reinfected animals (closed symbol).
(TIF)

**S5 Fig. Ultrasound measurements over time of Biparietal Diameter (BPD) and Femur Length (FL).** (A) BPD of CMV-seropositive reinfected (green) and CMV-seropositive controls (grey) fetuses compared to reference values [37]. (B) FL of CMV-seropositive reinfected

(green) and CMV-seropositive control (grey) fetuses compared to reference values [37].
(TIF)

**S1 Table. Animal details for study groups outlined in Table 1.**
(DOCX)

**S2 Table. RhCMV DNA PCR in placental and fetal tissues of CD4+ T lymphocyte-depleted dams.**
(DOCX)

**S3 Table. Antibodies for phenotyping and intracellular cytokine staining.**
(DOCX)

**S1 Data. Excel spreadsheet containing in separate sheets, the numerical raw data for Figs 1B, 1C, 2A, 2B, 3, 4A, 4B, 4C, 5A, 5B–5D, 6A, 6B, 7A, 7B, 7C and 7D.**
(XLSX)

## Acknowledgments

The Anti-CD4 [CDR1] antibody used in this study was provided by the NIH Nonhuman Primate Reagent Resource (P40 OD028116). The authors gratefully acknowledge the Veterinary Medicine staff at the Tulane National Primate Reseach Center (TNPRC) for care of the animals, TNPRC Flow Cytometry Core for acquisition of flow cytometry data and TNPRC Pathogen Detection and Quantification Core for real time PCR runs and Luminex assays.

## Author Contributions

**Conceptualization:** Peter Barry, Klaus Früh, Sallie R. Permar, Amitinder Kaur.

**Data curation:** Matilda J. Moström, Shan Yu, Dollnovan Tran, Cyril J. Versoza, Anne Mirza, Sarah Valencia, Jeffrey D. Jensen, Susanne P. Pfeifer, Timothy F. Kowalik, Amitinder Kaur.

**Formal analysis:** Matilda J. Moström, Shan Yu, Dollnovan Tran, Frances M. Saccoccio, Cyril J. Versoza, Scott Hansen, Jeffrey D. Jensen, Susanne P. Pfeifer, Timothy F. Kowalik, Amitinder Kaur.

**Funding acquisition:** Sallie R. Permar, Amitinder Kaur.

**Investigation:** Matilda J. Moström, Shan Yu, Sallie R. Permar, Amitinder Kaur.

**Methodology:** Matilda J. Moström, Shan Yu, Dollnovan Tran, Frances M. Saccoccio, Cyril J. Versoza, Anne Mirza, Sarah Valencia, Margaret Gilbert, Scott Hansen, Jeffrey D. Jensen, Susanne P. Pfeifer, Timothy F. Kowalik, Amitinder Kaur.

**Project administration:** Matilda J. Moström, Shan Yu, Dollnovan Tran, Sallie R. Permar, Amitinder Kaur.

**Resources:** Daniel Malouli, Margaret Gilbert, Robert V. Blair, Peter Barry, Klaus Früh, Jeffrey D. Jensen, Susanne P. Pfeifer, Timothy F. Kowalik, Sallie R. Permar, Amitinder Kaur.

**Software:** Amitinder Kaur.

**Supervision:** Dollnovan Tran, Timothy F. Kowalik, Sallie R. Permar, Amitinder Kaur.

**Validation:** Matilda J. Moström, Shan Yu, Amitinder Kaur.

**Visualization:** Matilda J. Moström, Frances M. Saccoccio, Scott Hansen, Amitinder Kaur.

**Writing – original draft:** Matilda J. Moström, Amitinder Kaur.

**Writing – review & editing:** Matilda J. Moström, Klaus Früh, Jeffrey D. Jensen, Susanne P. Pfeifer, Timothy F. Kowalik, Sallie R. Permar, Amitinder Kaur.

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
