## [Decision Letter · Decision Letter 0]

5 Jun 2023

Dear Dr. Kaur,

Thank you very much for submitting your manuscript "Protective effect of pre-existing natural immunity in a nonhuman primate reinfection model of congenital cytomegalovirus infection" for consideration at PLOS Pathogens. As with all papers reviewed by the journal, your manuscript was reviewed by members of the editorial board and by several independent reviewers. The reviewers appreciated the attention to an important topic. Based on the reviews, we are likely to accept this manuscript for publication, providing that you modify the manuscript according to the review recommendations.

The reviewers suggest some modifications to the text. We are returning the manuscript so these modifications can be made.

Sincerely,

Robert F. Kalejta

Academic Editor

PLOS Pathogens

Patrick Hearing

Section Editor

PLOS Pathogens

Kasturi Haldar

Editor-in-Chief

PLOS Pathogens

orcid.org/0000-0001-5065-158X

Michael Malim

Editor-in-Chief

PLOS Pathogens

orcid.org/0000-0002-7699-2064

The reviewers suggest some modifications to the text. We are returning the manuscript so these modifications can be made.

Reviewer Comments (if any, and for reference):

Reviewer's Responses to Questions

**Part I - Summary**

Reviewer #1: In this study, the authors developed a nonhuman primate model of congenital cytomegalovirus (cCMV) infection in rhesus macaques (Rh) to investigate the protective role of pre-existing maternal immunity against cCMV infection. They re-infected CD4+ T lymphocyte-depleted RhCMV-seropositive dams in late first trimester gestation with RhCMV, and reported activation of circulating monocytes, boosting of the pre-existing RhCMV-specific CD8+ T lymphocyte and antibody response in the reinfected dams but not in control CD4+ T lymphocyte-depleted dams. Placental transmission was detected in only one of five reinfected dams and there were no adverse fetal sequelae. Viral genomic analysis confirmed transmission of reinfected RhCMV strains across the placenta. The authors concluded that the reduction of placental transmission and absence of fetal loss after non-primary as compared to previously reported primary infection in CD4+ T lymphocyte-depleted dams indicates a protective role of preconception maternal CMV-specific CD8+ T lymphocyte and/or humoral immunity against cCMV infection.

Overall, this new NHP model of CMV will provide a valuable tool to the field for investigating the role and mechanism of pre-existing maternal immunity in cCMV. The results showing reduced placental transmission and no fetal loss after non-primary infection demonstrate a clear protective role of the pre-existing maternal immunity in this model. There are a few minor concerns. The group sizes of this study are small. Based on their results the authors suggest that CMV re-infection is more likely to contribute to placental transmission than reactivation. While none of the three control animals experienced placental transmission, only one of the five re-infected animals showed placental transmission, so it is unclear if there is a statistical difference. The reinfection study also used a relatively high dose of viral inoculum (2 x 10E6), which could skew cCMV results towards re-infection rather than re-activation. Finally, it would be helpful for the authors to clarify or reconcile their data with observations that in the human population, roughly 50% are CMV sero-positive (which is hypothesized to be protective based on this study) and yet still accountable for roughly 50% of cCMV cases. These minor concerns do not diminish the value of this excellent work but some clarifications or discussions in the text will be helpful.

Reviewer #2: In this interesting manuscript by Mostrom et al., Protective effect of pre-existing natural immunity in a nonhuman primate reinfection model of congenital cytomegalovirus infection, the authors examine a question that is of considerable interest to pediatricians, vaccinologists, and public health officials: namely, does pre-conception immunity to CMV provide (at least some) level of protection to the developing fetus against congenital CMV infection and its attendant morbidity?

In general the paper is very well conceived and well written. The experimental design is logical and informative. Appropriate controls are included. The authors leverage a nonhuman primate model of cCMV in rhesus macaques where 100% placental transmission and 83% fetal loss were previously seen in CD4+ T lymphocyte-depleted monkeys after primary RhCMV infection. The only caveat was that experimental numbers were small, but to be expected given the expense of these experiments. Be that as it may, to further investigate the protective effect of preconception maternal immunity, this study consisted of a reinfection in CD4+ T lymphocyte-depleted RhCMV-seropositive dams inoculated in late first or early second trimester. A variety of strains were used, including RhCMV strain 180.92 (n=2), a more wild-type strain of RhCMV that contains the homolog of UL128; RhCMV UCD52, and FL-RhCMVΔRh13.1/SIVgag, a wild type-like RhCMV clone with SIVgag inserted as an immunological marker (n=3).

One thing I would like to see the others do is comment more about the genotype and phenotype correlations with 180.92 and UCD52. This has been published in the past, but in fairness to the reader the authors should plainly and simply state the differences in these strains, both genotypically and phenotypically, from past work. This reviewer would invite the authors to plainly and simply state which strain is more like a wild-type strain that would be encountered in a pregnant patient in the human context.

The authors should also make it more clear (if indeed it’s true) that the experiment included a “mixing” experiment in which UCD52 was mixed with the FL-RhCMVΔRh13.1/SIVgag construct. This is implied in the abstract, but not stated clearly in the abstract. It should be stated with absolute clarity.

Moving on to the data, the experiments clearly support the conclusion that there is an early transient increase in circulating monocytes followed by boosting of the pre-existing RhCMV-specific CD8+ T lymphocyte and antibody responses observed in reinfected dams, but not in control CD4+ T lymphocyte-depleted dams. These results extend previous observations from the CD4+ depletion studies in this model and are informative. The demonstration of the emergence of SIV Gag-specific CD8+ T lymphocyte responses in macaques inoculated with the FL-RhCMVΔRh13.1/SIVgag virus variant is a useful and informative control confirming that reinfection took place.

The heart of the matter is that placental transmission was detected in only one of five reinfected dams and there were no adverse fetal sequelae. The authors make a rather large leap of faith, but probably an appropriate one, that this model provides good evidence that, in the human context, that even if re-infections occur, they are less likely to produce placental transmission, and unlikely to lead to sequelae if they occur, in CD4+ depleted dams in the reinfection setting. They conclude that reduced placental transmission and absence of fetal loss after non-primary as opposed to primary infection in CD4+ depleted dams indicates that preconception maternal CMV-specific CD8+ T lymphocyte and/or humoral immunity can protect against cCMV infection in this reinfection setting. There are two caveats to consider. One is the small number of monkeys studied. Even if reinfection leads to transmission associated-sequelae in human infants, the percentages are small. It would be an interesting exercise to calculate (although this reviewer doesn’t think the authors need to do this) to statistically surmise the “number of monkeys needed to treat” to see if sequelae were modified by pre-existing monkey immunity, if numbers were analogous to the human situation. What is the statistical power based on these small numbers?

The second point circles back to the strain variation question. Transmission of two reinfection RhCMV strains across the placenta is noted, with ~30% corresponding to RhCMVΔRh13.1/SIVgag and ~70% to RhCMV UCD52. Again, please clarify is this was a “mixing” challenge study and state the rationale for “mixing” (which does not occur, at least not to this extent, in nature). The inclusion of the SIVgag control is obvious, and useful (as noted above in this critique) but are there different levels of “fitness” for these viruses? Past work shows that UCD52 is wild-type in the pentameric complex region, in contrast to strain 68-1. But strain 180.92 is supposed to be UL128 positive, although it did undergo a lot of cell culture passage, both in rhesus and, interestingly, human cells (10.1128/JVI.80.8.4179-4182.2006). Page 7 states that the choices of inoculum strains were based on “past studies”. But the sequencing data does not mention any vertical transmission of 180.92 – only UCD52 and the derivative ΔRh13.1/SIVgag. Is this just because there was only two animals challenged (Table 1)? Or does UCD52 differ from 180.92 in terms of its transplacental transfer potential? If so, what is the molecular basis?

Reviewer #3: in this manuscript the authors investigate the role of pre-existing immunity in protection from congenital CMV (cCMV) in a RhCMV model. The authors previously reported 100% cCMV transmission and 83% fetal loss in CD4+ T lymphocyte-depleted CMV-seronegative rhesus macaque dams by intravenous delivery of mixed RhCMV viruses. Using a similar approach, the authors conducted the current study in CMV-seropositive rhesus macaques and report that pre-existing immunity protected cCMV, with 20% cCMV transmission and no fetal loss. Furthermore, virological and immunological characterizations supported that such protection against cCMV is most likely against reinfection of the inoculum viruses rather than reactivation. Overall, the study addressed a very important question in a NHP model that pre-existing immunity protects rhesus macaque dams from transmitting CMV to the fetus and from fetal loss even in the absence of CD4+ T cell response. However, there are few points that the authors should address before publication.

**Part II – Major Issues: Key Experiments Required for Acceptance**

Reviewer #1: (No Response)

Reviewer #2: No concerns. The experiments are well designed, have appropriate controls, and the conclusions are supported by the data.

Reviewer #3: The major limitation of the study is the use of the historical data from a previous study in CMV-seronegative rhesus macaques. This would be somewhat offset if the exact challenge condition was used. However, the inoculum virus and dose were lower and maybe less pathogenic when compared to the condition used in the historical study:

1a. Historical study (Nelson et al. 2017 JCI Insights) used a mixture of virus stocks: 2e6 180.92+1e6 UCD52+1e6 UCD59 (all in TCID50 units). This study used 2e6 180.92 only, or 1e6 UCD52+1e6 FL-RhCMVΔRh13.1/SIVgag, thus the challenge dose is reduced by roughly 2-fold.

1b. A separate historical study (Bialas et al. 2015 PNAS) has shown that inoculum of 180.92 alone resulted in more than 10-fold lower DNAemia than mixed viruses (180.92+UCD52+UCD59). In the latter group (challenge with mixed viruses), UCD52 was dominant among the three viruses in transmitting to fetus. Both data point to the direction that 180.92 is less pathogenic in terms of cCMV transmission, which was used for 2 out of 5 NHPs in the critical CMV-seropositive T cell depleted group in this study.

1c. Furthermore, 2 out of 5 NHPs in the critical CMV-seropositive T cell depleted group did not have an ideal depletion, when compared to the control group and the historical controls.

1d. Therefore, these three major differences on key experimental parameters may decrease confidence of the major conclusion of this study.

**Part III – Minor Issues: Editorial and Data Presentation Modifications**

Reviewer #1: Fig 1A: It will be helpful if the authors can clarify the rationale for using both RhCMV strain 180.92 and UCD52 in this study, and also which animal was reinfected with which virus strain. The information is somewhere there but it will be better to have this at the beginning.

Fig 1B: the labels of individual curves are confusing, particularly JP01 vs. 292-09, and 274-05 vs. KK24

Fig 2B: As stated above, the group sizes of this study are small (i.e. 1/5 of reinfected animals vs. 0/3 of control animals were transmitters) and the re-infection dose is high so that it is difficult to draw a firm conclusion as stated on Line 177: “Taken together, these data demonstrate placental transmission after RhCMV reinfection but not after reactivation”. The authors should clearly state the caveats of the study.

Fig 4B: Please clarify what the size of the pier chart stands for. For instance, in 4C, 3-functional T cells are shown as 10% but in 4B, it appears to be 75% of the pier chart.

Fig 5A: It appears that the colors of CD4+ T cell and LogED50 curves are mislabeled for 274-05.

Fig 5C: please clarify if these are folds of the increase.

Line 227: The authors stated that “our results demonstrate elevation of RhCMV-specific CD8+ T lymphocyte and antibody responses that was evident only after RhCMV reinfection, not reactivation following CD4+ T lymphocyte depletion”. Again one caveat here is that the inoculation dose is high, which could skew the result towards re-infection.

Line 247: The authors stated that “the resulting whole genome sequencing data (mean read length: ~200 bp) was mapped against the RhCMV UCD52 and FL-RhCMVΔRh13.1/SIVgag reference assemblies”. Please clarify if these reads are unique to these two viral strains.

Reviewer #2: Consider expanding Table 1 to include columns describing in more details the differences, both genotypic and phenotypic, of strains 180.92 and UCD52.

Reviewer #3: 2. Some key figure legends, text descriptions, and assay result lack consistency and technical rigor:

2a. Figure 1B: Two sets of two NHPs share the same label.

2b. Figure 2A: Which two NHPS received only 180.92?

2c. Figure 2 & 7A: How LOD of 1 copy per mL was achieved for plasma and amniotic fluid samples? What’s the real LOD?

2d. Figure 3: Please explain the rationale that only 3 out of 5 NHPs are described in this assessment?

2e. Figure 4 legend described that 4 out of NHPs were infected by UCD52+ FL-RhCMVΔRh13.1/SIVgag, inconsistent from the text.

2f. Figure 5A: The same CD4 T cell count/uL data was presented as Figure 1B. However, the data are very different between the two figures.

2j. Figure 5A ELISA data: 274-05 does not have a 0.8 log increase of ED50 as labeled.

2h. Figure 6A: Please include the same data for JP01 and especially KK24 which had the cCMV case.

3. Figure 3: please explain the drop of %B cells upon CD4 T cell depletion

4. Table 1: It will be nice to include to table 1 or a supplemental table about dates of depletion/challenge, center/location, virus stock#/dose (same virus stock used for all studies?), so the reader can appreciate the potential differences among studies and animals.

PLOS authors have the option to publish the peer review history of their article (what does this mean?). If published, this will include your full peer review and any attached files.

Reviewer #1: No

Reviewer #2: No

Reviewer #3: No

Figure Files:

Data Requirements:

Reproducibility:

References:

---

## [Editor Report · Decision Letter 1]

29 Aug 2023

Dear Dr. Kaur,

We are pleased to inform you that your manuscript 'Protective effect of pre-existing natural immunity in a nonhuman primate reinfection model of congenital cytomegalovirus infection' has been provisionally accepted for publication in PLOS Pathogens.

Best regards,

Robert F. Kalejta

Academic Editor

PLOS Pathogens

Patrick Hearing

Section Editor

PLOS Pathogens

Kasturi Haldar

Editor-in-Chief

PLOS Pathogens

orcid.org/0000-0001-5065-158X

Michael Malim

Editor-in-Chief

PLOS Pathogens

orcid.org/0000-0002-7699-2064
---

## [Editor Report · Acceptance letter]

13 Sep 2023

Dear Dr. Kaur,

We are delighted to inform you that your manuscript, "Protective effect of pre-existing natural immunity in a nonhuman primate reinfection model of congenital cytomegalovirus infection," has been formally accepted for publication in PLOS Pathogens.

Best regards,

Kasturi Haldar

Editor-in-Chief

PLOS Pathogens

orcid.org/0000-0001-5065-158X

Michael Malim

Editor-in-Chief

PLOS Pathogens

orcid.org/0000-0002-7699-2064